palaeontology

Bissekty Formation, Carcharodontosauria, dinosaur, Late Cretaceous, Tyrannosauroidea, Uzbekistan

**Author for correspondence:**
Kohei Tanaka
e-mail: koheitanaka@geol.tsukuba.ac.jp

# A new carcharodontosaurian theropod dinosaur occupies apex predator niche in the early Late Cretaceous of Uzbekistan

Kohei Tanaka[1], Otabek Ulugbek Ogli Anvarov[1], Darla K. Zelenitsky[2], Akhmadjon Shayakubovich Ahmedshaev[3] and Yoshitsugu Kobayashi[4]

[1]Life and Environmental Sciences, University of Tsukuba, Tsukuba 305-8572, Ibaraki, Japan
[2]Department of Geoscience, University of Calgary, Calgary, Alberta, Canada T2N 1N4
[3]State Geological Museum of the State Committee of the Republic of Uzbekistan on Geology and Mineral Resources, Tashkent 100060, Republic of Uzbekistan
[4]Hokkaido University Museum, Hokkaido University, Sapporo 060-0810, Hokkaido, Japan

KT, 0000-0003-4721-6418

Carcharodontosauria is a group of medium to large-sized predatory theropods, distributed worldwide during the Cretaceous. These theropods were probably the apex predators of Asiamerica in the early Late Cretaceous prior to the ascent of tyrannosaurids, although few Laurasian species are known from this time due to a poor rock record. Here, we describe *Ulughbegsaurus uzbekistanensis* gen. et sp. nov. from the early Late Cretaceous (Turonian) of Central Asia, which represents the first record of a Late Cretaceous carcharodontosaurian from the region. This new taxon is represented by a large, isolated maxilla from the Bissekty Formation of the Kyzylkum Desert, the Republic of Uzbekistan, a formation yielding a rich and diverse assemblage of dinosaurs and other vertebrates from fragmentary remains. Comparison of the maxilla with that of other allosauroids indicates *Ulughbegsaurus* was 7.5–8 m in body length and greater than 1000 kg in body mass, suggesting it was the previously unrecognized apex predator of the Bissekty ecosystem while smaller known tryannosauroids and dromaeosaurids were probable mesopredators. The discovery of *Ulughbegsaurus* records the geologically latest stratigraphic co-occurrence of carcharodontosaurid and tyrannosauroid dinosaurs from Laurasia, and evidence indicates carcharodontosaurians remained the dominant predators relative to tyrannosauroids, at least in Asia, as late as the Turonian.

# 1. Introduction

Allosauroids were globally distributed medium to large predatory theropods that dominated various ecosystems during the Middle Jurassic to the end Cretaceous. The latest surviving group was Carcharodontosauria, which thrived in Gondwana from the Late Jurassic to the end of the Cretaceous (e.g. [1,2]), but disappeared in Laurasia after the Turonian. Carcharodontosaurians attained extreme body sizes (greater than 6000 kg [3]), rivalling those of tyrannosaurids and spinosaurids. Considered as the apex predators in Laurasia until the mid-Cretaceous [1,4], carcharodontosaurians and other large-bodied theropods (ceratosaurids and spinosaurids) disappeared after which tyrannosauroids (primarily tyrannosaurids) occupied top predatory niches during the last 20 million years of the Cretaceous (Campanian and Maastrichtian).

The disappearance of carcharodontosaurians and other allosauroids at the end Turonian is probably related to the ascent of tyrannosauroids, as tyrannosaurids, in Late Cretaceous ecosystems of Asiamerica [5]. However, the transition of these apex predator faunas is poorly understood due to a paucity of pertinent lower Upper Cretaceous deposits and a scarcity of theropod remains from them [5,6]. Despite this, species of carcharodontosaurians have been recovered from two lower Upper Cretaceous formations (Cenomanian and Turonian) of Asiamerica and include large taxa [5,6]. As the apex predators in these early Late Cretaceous ecosystems, their faunal associations with other predators, particularly tyrannosauroids, are poorly known. The only Late Cretaceous Laurasian occurrence of a carcharodontosaurian and a tyrannosauroid species in the same formation is from the Cenomanian (98 Ma) of North America (*Siats* and *Moros*, respectively), a co-occurrence that has been used to suggest that large carcharodontosaurians were excluding tyrannosauroids from apex predatory niches [4,6]. In Asia, such a co-occurrence of these two clades is only known from the Upper Jurassic Shishugou Formation (*Sinraptor* and *Guanlong* [7–9]), a time period well before the ascent of tyrannosauroids.

In this paper, we describe a new Late Cretaceous (Turonian) specimen of a relatively large-bodied carcharodontosaurian, represented by an isolated maxilla, from the Bissekty Formation of the Republic of Uzbekistan, Central Asia (electronic supplementary material, figure S1). Anatomical terminology follows Hendrickx & Mateus [10] and Hendrickx *et al.* [11]. This well-studied formation has yielded a rich and diverse ecosystem with numerous dinosaurs (including Hadrosauroidea, Ceratopsia, Sauropodomorpha, Ornithomimidae, Dromaeosauridae and Tyrannosauroidea) and other vertebrates (e.g. [12–19]). Our discovery adds to few known Late Cretaceous carcharodontosaurian species from Asiamerica and reveals a previously unknown apex predator among mid-sized predatory tyrannosauroids and dromaeosaurids in the Turonian Bissekty ecosystem.

Institutional abbreviations: CCMGE, Chernyshev's Central Museum of Geological Exploration, Saint Petersburg, Russia; UzSGM, State Geological Museum of the State Committee of the Republic of Uzbekistan on Geology and Mineral Resources, Tashkent, Uzbekistan; ZIN PH, Paleontological Collection, Zoological Institute, Russian Academy of Science, Saint Petersburg, Russia.

# 2. Systematic palaeontology

Dinosauria Owen, 1842 [20]
Saurischia Seeley, 1887 [21]
Theropoda Marsh, 1881 [22]
Tetanurae Gauthier, 1986 [23]
Allosauroidea Marsh, 1878 [24]
Carcharodontosauria Benson *et al.*, 2010 [1]
  *Ulughbegsaurus uzbekistanensis* gen. et sp. nov.
Zoobank ID: urn:lsid:zoobank.org:pub:A468DE62-F159-4569-898D-89A232A88492 (for this publication), urn:lsid:zoobank.org:act:854798B5-B4E0-492D-B5F9-0267FC79E2F6 (for the new genus) and urn:lsid:zoobank.org:act:96749FD7-30BF-4322-8434-6C85DAC23105 (for the new species)
Holotype: UzSGM 11-01-02, a left maxilla.
Etymology: '*Ulughbeg*' refers to Timurid sultan Ulugh Beg, in recognition of his early scientific contributions as a fifteenth-century astronomer and mathematician in central Asia region (now Uzbekistan). '*Sauros*' meaning reptiles in Latin. Specific name, '*uzbekistan*', refers to the Republic of Uzbekistan.

Referred specimens: CCMGE 600/12457 is a jugal ramus of a left maxilla and ZIN PH 357/16 is a posterior end of the right maxilla, previously referred to the dromaeosaurid *Itemirus medullaris* ([18]: see '5. Comments on isolated teeth and bone fragments of theropods from the Bissekty Formation').

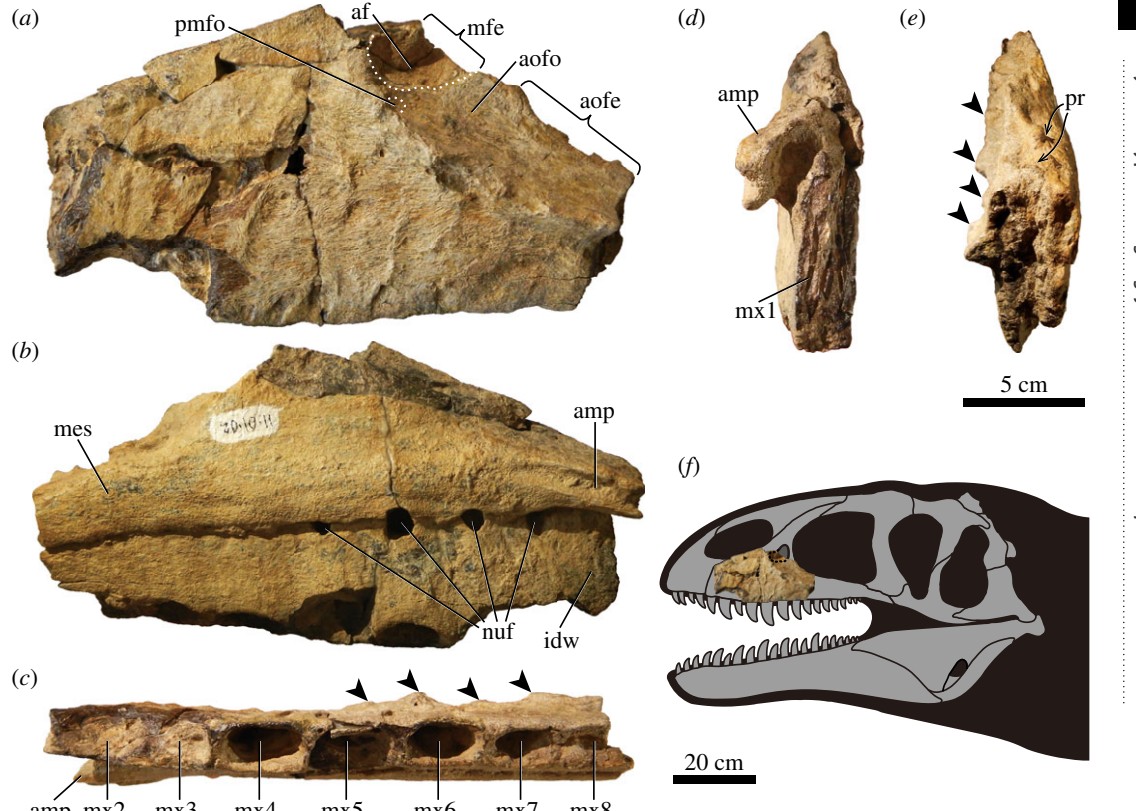

**Figure 1.** Left maxilla of *Ulughbegsaurus* (UzSGM 11-01-02). (*a*) Lateral, (*b*) medial, (*c*) ventral, (*d*) anterior and (*e*) posterior views. (*f*) Reconstruction of skull (grey missing bones are based on *Neovenator*, modified from Naish *et al.* [25]). Black arrows indicate tubercles on the rim of the antorbital fossa and dashed lines indicate depressions. Abbreviations: af, accessory fossa; amp, anteromedial process; aofe, antorbital fenestra; aofo, antorbital fossa; idw, interdental wall; mes, medial shelf; mfe, maxillary fenestra; mx1-8, first to eighth maxillary tooth; nuf, nutrient foramen; pmfo, promaxillary fossa; pr, pneumatic recess.

Locality and horizon: The holotype and referred specimens were recovered at Dzharakuduk, central Kyzylkum Desert, Navoi Viloyat, Uzbekistan, where the Upper Cretaceous (middle to upper Turonian, *ca* 90–92 million years ago [19]) Bissekty Formation is exposed; the precise location is unknown as geographic coordinates were not noted in the field. The holotype was brought to UzSGM by a field team member of Lev Alexandrovich Nessov.

Diagnosis: Carcharodontosaurian theropod with a series of shallow, oval depressions on the lateral surface along the ventral edge of maxilla; tubercles along with the ridged rim of the antorbital fossa; vertically oriented ridges on the lateral surface of maxilla and large foramina at a dorsal edge of dental plates in maxilla.

## 3. Description and comparisons

UzSGM 11-01-02 is a left maxilla consisting of the main body (i.e. maxillary body) and the anterior half of the jugal ramus. The entire ascending ramus is missing (figure 1). The maxilla as preserved is 24.2 cm in length (anteroposterior) and 13.1 cm in height (dorsoventral). The contact surface for the premaxilla is not preserved due to damage at the anterior margin of the maxilla, where the first alveolus is broken and exposes an implanted tooth.

The maxillary body is subtriangular in a lateral view (figure 1*a*). The anterodorsal margin is nearly straight and contributes to the external naris as in *Allosaurus* [26] and *Neovenator* [26,27], but unlike *Acrocanthosaurus* [28]. Judging from the angle of the maxillary tooth in the first alveolus, the contact surface for the premaxilla at the anterior margin of the maxilla is presumably subvertical as in *Allosaurus* [29], *Neovenator* [27] and *Yangchuanosaurus* [30]. This condition differs from the posterodorsally inclined margin of Carcharodontosauridae [5,26,27,31–34] and *Sinraptor* [7,35].

The lateral surface of the maxilla has a rugose texture as reported in many members of Carcharodontosauria [27,32,34,36]. Additional surface texturing of the maxilla consists of at least six

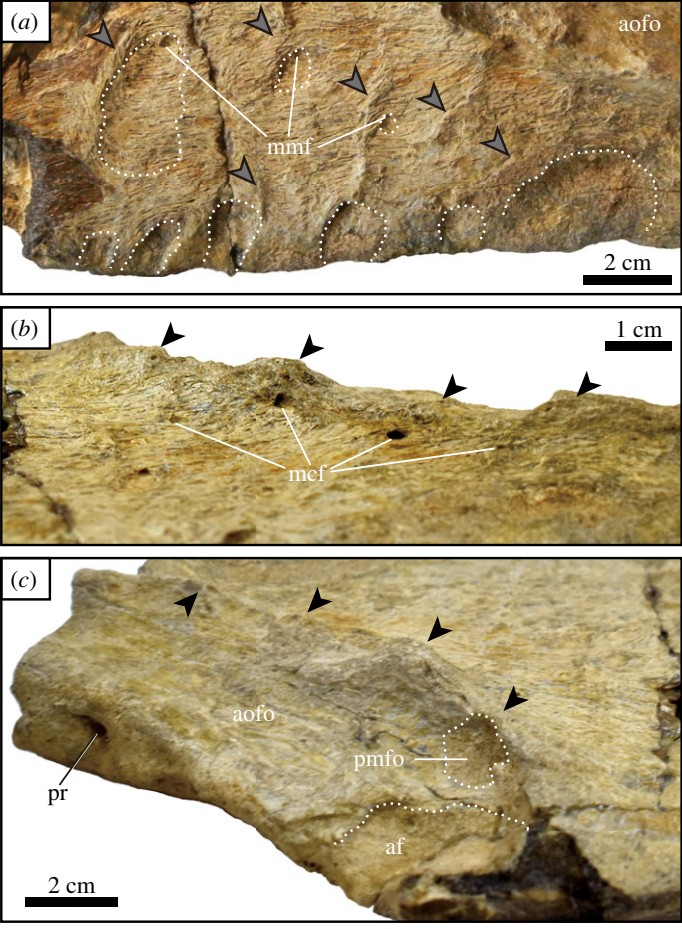

**Figure 2.** Close-up views of the left maxilla (UzSGM 11-01-02). (*a*) Lateral surface of the main body near ventral margin, (*b*) the rim of the antorbital fossa in ventrolateral view and (*c*) dorsolateral view of the antorbital fossa. Black arrows indicate tubercles on the rim of the antorbital fossa, grey arrows indicate dorsoventrally oriented ridges, and dashed lines indicate depressions. Abbreviations: af, accessory fossa; aofo, antorbital fossa; mcf, maxillary circumfenestra foramen; mmf, maxillary median foramen; pmfo, promaxillary fossa; pr, pneumatic recess.

dorsoventrally extending ridges that are situated between the second and seventh alveoli (figure 2*a*). These ridges are comparable to those in some tyrannosaurids (e.g. *Tarbosaurus*, *Thanatotheristes*, *Tyrannosaurus* and *Zhuchengtyrannus* [37–39]). The lateral surface of the maxilla also bears a number of shallow depressions that are often bounded by a ridge similar to some large tyrannosaurids (figure 2*a* [37,38]). Three more dorsally positioned depressions are each pierced by a single, minute maxillary median foramen at the dorsal edge (figure 2*a*). A series of shallow, oval depressions are also visible along the ventral margin of the maxilla (figure 2*a*).

The rim of the antorbital fossa is gently curved, similar to other allosauroids [5,7,27,29,33] and is rugose and ornamented with tubercles (figures 1*c,e* and 2*b,c*). A series of tiny maxillary circumfenestra foramina are located just anteroventral to the rim (figure 2*b*). An oval accessory fossa (39 mm in diameter) is found in the anterior-most preserved portion of the antorbital fossa, which is bounded posteriorly by the anterior margin of the maxillary fenestra (figures 1*a* and 2*c*). This accessory fossa is considered here to be an extension of the maxillary fenestra due to its adjacent location. Another small fossa (18 mm in diameter), presumably a promaxillary fossa, is present at the anteroventral corner of the antorbital fossa, just anteroventral to the accessory fossa (figure 2*c*). This small promaxillary fossa is obscured laterally by the rim of the antorbital fossa, similar to that of *Allosaurus*, *Neovenator* and *Sinraptor* [27]. Just medial to the margin of the antorbital fenestra, the dorsal surface of the jugal ramus has anteromedial and ventromedial pneumatic recesses (figure 1*e*).

On the medial side, the anteromedial process of the maxilla is well preserved (4 cm in anteroposterior length) and extends nearly horizontally (figure 1*b,d*). A single horizontal groove on the medial surface of

this process is potentially a contact for the vomer, premaxilla and the opposite bone. The dorsal surface of the process to the apex of the main body represents a contact surface for the nasal. This surface is flat, smooth and narrows posteriorly. The medial shelf extends posteriorly on the main body from the anteromedial process. The surface of the maxilla dorsal to the medial shelf is flat and smooth, whereas the interdental wall below, consisting of interdental plates, is slightly rugose, as is seen in other Carcharodontosauria [5,27,33,36,40]. As in most allosauroids [26], the interdental plates are fused together. Although the interdental wall is broken ventrally along with the anterior alveoli, it is deep where well preserved (47 mm at the fourth alveolus), but becoming shallow posteriorly (31 mm at the eighth alveolus). A series of four large nutrient foramina is present at the dorsal margin of the interdental wall below the medial shelf (figure 1b) in which the foramen between the fourth and fifth alveoli is particularly large (14 mm in diameter).

The first eight sub-rectangular alveoli are preserved (figure 1c,d). An unerupted tooth remains in the fifth alveolus. Of the measurable alveoli, the fifth is the largest (33.3 × 16.6 mm) after which the alveoli decrease in size posteriorly. Alveolus size, on average, is most comparable to that of *Neovenator* and *Shaochilong* (electronic supplementary material, table S1 and figure S2).

# 4. Phylogenetic analysis

Two phylogenetic analyses, each including the new taxon *Ulughbegsaurus*, were conducted with the software TNT v. 1.5 [41] (electronic supplementary material, text S1: figures S3 and S4). Our first analysis was performed using the data matrix proposed by Carrano *et al*. [42] then modified by Hendrickx & Mateus [10] where *Eoraptor* represents the outgroup; the characters are treated as unordered. Our second analysis was conducted using the matrix originally proposed by Porfiri *et al*. [43] then modified by Chokchaloemwong *et al*. [40] where *Ceratosaurus* represents the outgroup; characters 2, 4, 6, 13, 15, 17, 27, 69, 106, 148, 155, 158, 160, 167, 169, 171, 179, 181, 194, 195, 205, 208, 217, 233, 241, 259, 267, 271 were treated as ordered. A traditional heuristic search was done with 1000 replicates of Wagner trees using random addition sequences, followed by the tree bisection and reconnection branch swapping that holds 10 trees per replicate. Consistency index and retention index were calculated with PAUP 4.0a [44].

The first phylogenetic analysis produced 6320 most parsimonious trees with a strict consensus tree placing *Ulughbegsaurus* within a poorly resolved Neovenatoridae (*Aerosteon*, *Australovenator*, *Chilantaisaurus*, *Fukuiraptor*, *Megaraptor* and *Neovenator*), a clade within Carcharodontosauria (figure 3a). Neovenatoridae was supported by 12 synapomorphies in this analysis, which includes two characters of the maxilla (i.e. small foramen of promaxillary fenestra and sculptured external surface of maxilla and nasal). The second phylogenetic analysis produced 284 most parsimonious trees with a strict consensus tree recovering *Ulughbegsaurus* within Carcharodontosauria where *Ulughbegsaurus*, *Siamraptor*, *Eocarcharia*, *Neovenator*, *Concavenator* and the clade of *Acrocanthosaurus*, *Shaochilong* and Carcharodontosaurinae all form a polytomy (figure 3b). Carcharodontosauria is supported by 18 synapomorphies, including two characters of the maxilla (i.e. fused posterior paradental plates and approximately 20° ventral orientation at the jugal contact). A major difference in the results of these two phylogenetic analyses is that Megaraptora is placed within Carcharodontosauria and Tyrannosauroidea in the first and second analyses, respectively. Our results are consistent with the results of the original analyses where Megaraptora is placed in Allosauroidea by Carrano *et al*. [42] but in Tyrannosauroidea by Porfiri *et al*. [43]. Based on both of our phylogenetic analyses, however, it is evident that *Ulughbegsaurus* is assignable to the Carcharodontosauria.

# 5. Comments on isolated teeth and bone fragments of theropods from the Bissekty Formation

Although difficult to demonstrate unequivocally at present, it is possible that some isolated teeth from the Bissekty Formation are attributable to carcharodontosaurians, based on alveoli measurements as a proxy for tooth dimensions. The maxillary alveoli of the *Ulughbegsaurus* individual are relatively large with mesiodistal lengths ranging from 21 to 33 mm (mean 28.6 mm, $n = 5$) (electronic supplementary material, table S1). Alveoli measured from a maxilla referable to *Timurlengia* are smaller (mesiodistal length: 19.3–21.7 mm, $n = 3$) (electronic supplementary material, table S1 and figure S5) than *Ulughbegsaurus*, as are those previously reported in a maxilla (largest is 19.5 mm) and dentary (largest is 22 mm in ZIN PH 15/16) [13] referred to *Timurlengia* [19]. Also, *Ulughbegsaurus* tends to have

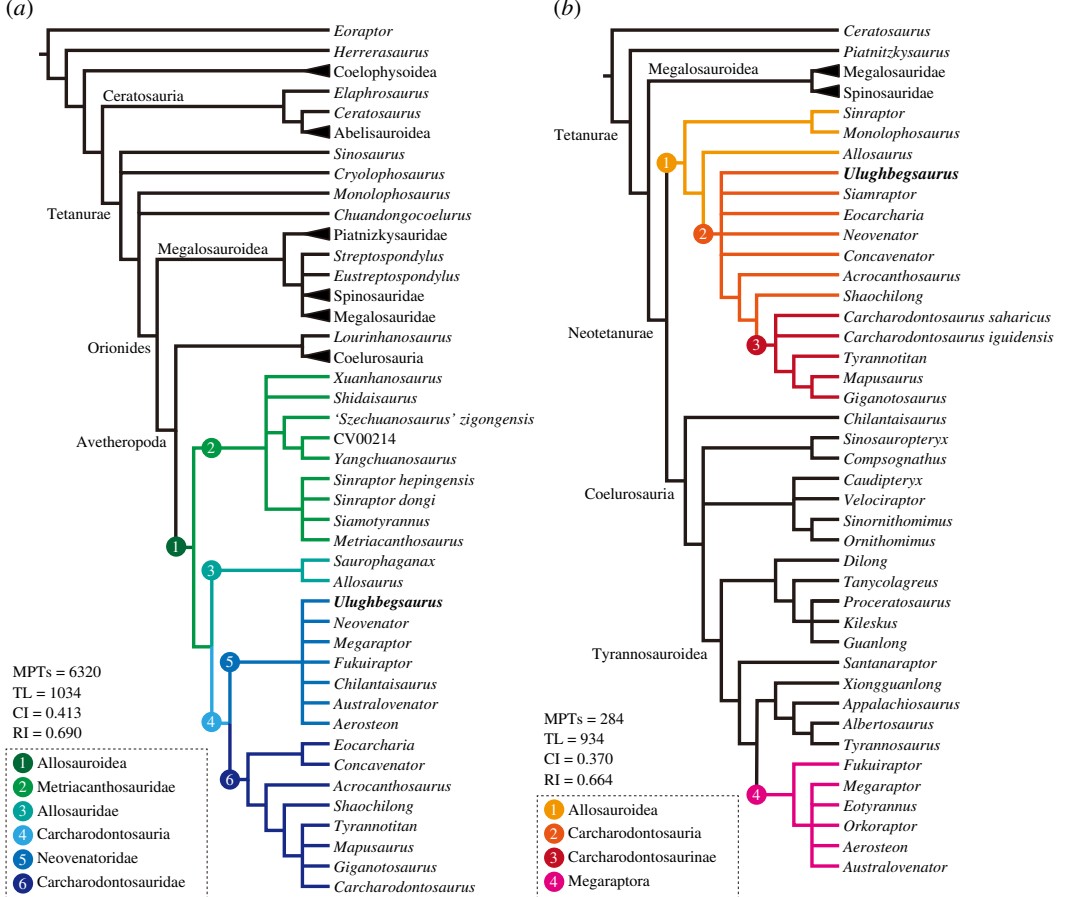

**Figure 3.** Simplified cladograms of theropod taxa, indicating the phylogenetic positions of *Ulughbegsaurus*. (*a*) Consensus tree based on the data matrix of Hendrickx & Mateus [10] and (*b*) consensus tree based on Chokchaloemwong *et al.* [40]. See electronic supplementary material, figures S3 and S4 for the complete cladograms. Abbreviations: CI, consistency index; MPTs, most parsimonious trees; RI, retention index; TL, tree length.

relatively narrow alveoli (ratio of labiolingual width and mesiodistal length: 0.48–0.54, mean 0.51 mm) compared with those of *Timurlengia* (0.51–0.74, mean 0.64), consistent with carcharodontosaurians in general, which have significantly narrower teeth than tyrannosauroids ($p < 0.01$, see electronic supplementary material, figure S6). However, overlap in both alveolus length and ratio values are seen between *Ulughbegsaurus* and *Timurlengia*. Many of several large, isolated theropod teeth (crown base length: CBL > 15 mm) from the Bissekty Formation (at UzSGM), are narrow with crown base length to crown base width ratios (CBR) values (0.45–0.56, mean 0.50, $n = 7$) similar to the alveolus ratios of *Ulughbegsaurus* (electronic supplementary material, figure S6). With the largest alveoli of *Timurlengia* at a mesiodistal length of 22 mm, it is possible that some of the large tyrannosauroid teeth reported from Dzharakuduk of the Bissekty Formation (with CBLs up to 26.6 mm and CBR to 0.41–0.70 [13]) could belong to carcharodontosaurians.

Some fragmentary bones reported from Dzharakuduk of the Bissekty Formation may also belong to carcharodontosaurians rather than to other theropods. The posterior portion (jugal ramus) from a large left maxilla (CCMGE 600/12457) was described initially as tyrannosauroid by Nessov [45], then redescribed by Sues & Averianov [18] and tentatively assigned to a dromaeosaurid *Itemirus medullaris* Kurzanov, 1976 [46]. Although the preserved portions of this specimen and UzSGM 11-01-02 do not overlap anatomically, both specimens are from a similarly large-sized bone (mediolateral thickness of the jugal ramus from both specimens approximately 23 mm [45]), both have fused interdental plates forming a solid interdental wall, and both have a series of small tubercles along the rim of the antorbital fossa (figure 3*a* of [18]). These similarities, and the tubercles in particular, which are an autapomorphy of *Ulughbegsaurus*, suggest this posterior maxillary fragment (CCMGE 600/12457) belongs to the carcharodontosaurian *Ulughbegsaurus*. Also, morphologically similar to CCMGE 600/12457, a smaller piece of the posterior end of the right maxilla (ZIN PH 357/16 [18]) without the antorbital fossa rim preserved could also belong to this carcharodontosaurian.

# 6. Body size estimation of *Ulughbegsaurus*

The size of the maxilla in theropods can be used as a proxy for body size [5]. It has been demonstrated in large theropods, such as tyrannosaurids [47], that the length of the maxillary tooth row is isometrically correlated with femur length, and femur length is widely used as an indicator of body mass [4,48,49]. For *Ulughbegsaurus,* the complete length of the maxillary tooth row is unknown for the holotype (UzSGM 11-01-02). However, the preserved portion of the maxillary tooth row from the second to the eighth alveoli is approximately 23 cm in length, which is at least about 20% longer than a relatively large (greater than 1000 kg) allosauroid, such as *Yangchuanosaurus* (the length from the second to the eighth alveoli 19.5 cm and total body length 7 m: electronic supplementary material, table S4). This measurement indicates that this *Ulughbegsaurus* individual was at least 7 m in total body length and over 1000 kg in body mass. Additional regression analyses using estimates of the maxilla length and maxillary tooth row length measurements, provided in electronic supplementary material, text S2 and figures S7 and S8, indicate it was 7.5–8.0 m in body length.

# 7. Discussion

Although carcharodontosaurians (includes Megaraptora in this discussion) were widespread in Laurasia during the Early to early Late Cretaceous, known from Europe (e.g. *Concavenator* and *Neovenator*), North America (e.g. *Acrocanthosaurus* and *Siats*), and East and Southeast Asia (e.g. *Fukuiraptor, Shaochilong* and *Siamraptor*) (e.g. [1,4,5,40,50]), *Ulughbegsaurus uzbekistanensis* gen. et sp. nov. represents the first definitive fossil evidence of carcharodontosaurians from Central Asia. Thus, its discovery fills a geographic gap in the clade between Europe and East Asia and shows that carcharodontosaurians were widespread across Asia. While carcharodontosaurians from East Asia have affinities with Gondwanan taxa [1,5], it remains ambiguous, due to unresolved relationships in our phylogenetic analyses (figure 3), if carcharodontosaurians dispersed to Central Asia from East Asia or from Europe. With Central Asia as the location of the European-Asian land connection that appeared during regression of Turgai sea, this area may have allowed dispersal of some dinosaur groups (e.g. ornithopods, sauropods and theropods) among regions of Laurasia [5].

Within the rich and diverse dinosaur fauna of the Bissekty Formation (e.g. [12–19]), our discovery of this predatory theropod *Ulughbegsaurus*, at 7.5–8.0 m in length and greater than 1000 kg in mass, probably occupied the role of apex predator in this Turonian ecosystem. Although the remains of several predatory theropods are described from the formation (*Itemirus medullaris, Paronychodon asiaticus, Richardoestesia asiatica, Timurlengia euotica* and *Urbacodon* sp. [12,14,15,17–19]), these taxa are smaller forms than *Ulughbegsaurus*. The one tyrannosauroid, *Timurlengia*, is only 3–4 m long with a body mass of about 170 kg ([19]: electronic supplementary material, table S2), which is comparable to the tyrannosauroid *Xiongguanlong* (Aptian–Albian [19]). *Ulughbegsaurus* is larger than this tyrannosauroid as the estimated maxilla size is nearly twice as long as that of *Timurlengia* ([13]: electronic supplementary material, figure S7). To date, an isolated large second pedal phalanx of a relatively large dromaeosaurid has been reported [18], an individual the authors estimated as comparable in size to *Utahraptor* (= *Achilobator* dromaeosaurid that is 5 m in total length and 350–450 kg in weight: [3,51]). Although precise body sizes are difficult to estimate from most isolated bones, a dromaeosaurid and a tyrannosauroid of similar size would have partitioned their roles as the two apex predators in this Turonian ecosystem [18]. However, *Ulughbegsaurus* (7.5–8.0 m in estimated length), belonging to a group of known large theropods, reveals a carcharodontosaurian was probably the apex predator of the Bissekty ecosystem rather than the smaller tyrannosauroids and dromaeosaurids that were mesopredators.

The discovery of *Ulughbegsaurus* adds to our understanding of the dinosaur-dominated ecosystems of early Late Cretaceous Laurasia, as one of few species of relatively large apex predators in Asia from this time (see [5]). This time interval also records a top predator (allosauroid-tyrannosauroid) faunal turnover in Asiamerica, although has yielded few large theropod species due to a poor rock record [5]. The Turonian Bissekty Formation is the first formation in the Late Cretaceous of Asia to yield a carcharodontosaurian species co-occurrence with a tyrannosauroid [19]. The only other such Late Cretaceous co-occurrence is a large carcharodontosaurian allosauroid (*Siats*) and a small tyrannosauroid (*Moros*), from an earlier time period of the USA, in the Cenomanian part of the Cedar Mountain Formation (approx. 98 Ma) [4,6]. Thus, the Bissekty Formation provides the geologically latest evidence of sympatry between an allosauroid and a tyrannosauroid species in Late Cretaceous Laurasia, although the body size difference between these two theropods is not as large as those of

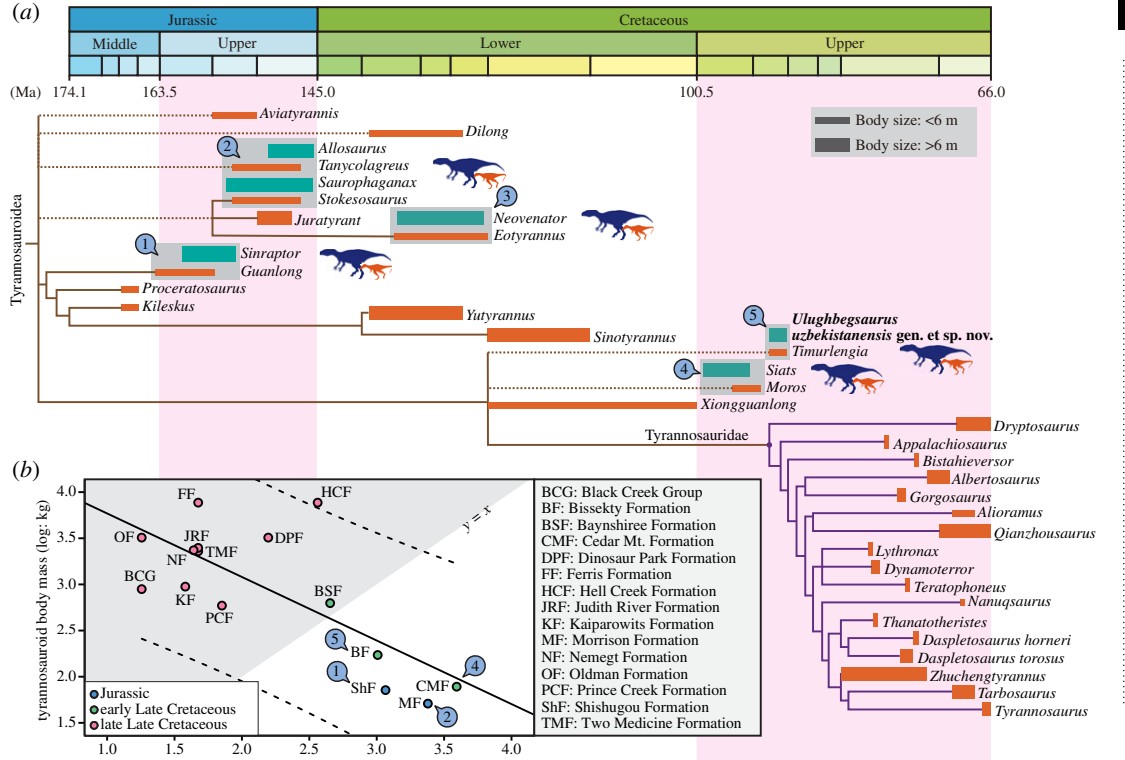

**Figure 4.** Comparisons between small tyrannosauroid and large non-tyrannosauroid predatory theropods. (*a*) Phylogenetic tree of Tyrannosauroidea with sympatric allosauroid taxa (1, *Guanlong* with sympatric *Sinraptor* from the Late Jurassic Shishugou Formation of China; 2, *Tanycolagreus* and *Stokesosaurus* with sympatric *Allosaurus* and *Saurophaganax* from the Late Jurassic Morrison Formation of the United States; 3, *Eotyrannus* and sympatric *Neovenator* from the Early Cretaceous Wessex Formation of the United Kingdom; 4, *Moros* and sympatric *Siats* from the early Late Cretaceous Cedar Mountain Formation of the United States; 5, *Timurlengia* and sympatric *Ulughbegsaurus* from the Turonian Bissekty Formation of Uzbekistan), indicating that sympatric large allosauroid taxa are found at least until Turonian. Phylogenetic tree is time calibrated based on Zanno *et al*. [6], Carr *et al*. [38] and Voris *et al*. [39], except for branch length and divergence times. (*b*) Bivariate plot of body mass between tyrannosauroids and non-tyrannosauroid predatory theropods that stratigraphically co-occur. Only the largest taxa were represented in each formation. The ordinary least-squares regression (solid line: $y = -0.689x + 4.458$) with the 95% confidence intervals (dashed lines) shows a negative correlation ($R^2 = 0.548$), indicating that tyrannosauroids were small when other large predatory theropods were present. The grey shadow is where tyrannosauroids are larger than non-tyrannosauroid theropods ($y > x$). Dataset for the bivariate plot is shown in the electronic supplementary material, table S2.

the Cenomanian formation. Thus, allosauroids were widespread as ecosystem large predators (see also [5]) in Asia, as late as the Turonian.

In the early Late Cretaceous and earlier, allosauroids were often the large apex predators in Asiamerica [1,4], even in formations where tyrannosauroid species co-occurred (electronic supplementary material, table S2). Body mass of tyrannosauroids relative to other sympatric predatory theropods (including allosauroids) shows a negative correlation, suggesting that tyrannosauroids were mesopredators or smaller predators of most ecosystems until achieving large body size in the Late Cretaceous (figure 4*b*). After the extinction of Laurasian carcharodontosaurians in the Turonian, substantive evidence of larger tyrannosauroids is not known until the Campanian of North America [52]. Although *Ulughbegsaurus* reveals that, relative to allosauroids, tyrannosauroids were mesopredators in the Turonian, there still was a gap (in the Coniacian–Santonian) where evidence of tyrannosauroid body size relative to other predatory theropods is lacking.

Among the mid-Cretaceous deposits in Asia, the faunal composition of the Bissekty Formation is most comparable to that of the Baynshiree Formation in eastern Mongolia, recently redated to late Cenomanian to Turonian [53]. Both formations produce Ankylosauria, Ceratopsia, Hadrosauroidea, Maniraptora (including giant dromaeosaurids [18,54]), Ornithomimosauria, Titanosauria and Tyrannosauroidea (electronic supplementary material, table S3), although Allosauroidea is currently unknown from the Baynshiree Formation. The similarities between dinosaur faunas of the Bissekty

and Baynshiree formations are of great interest because their respective areas were located at east–west extremes of the continent relative to one another. These similarities may imply that terrestrial vertebrates were relatively uniform in Turonian Laurasia, supporting the cosmopolitanism of dinosaur faunas in mid-Cretaceous.

# 8. Conclusion

Although carcharodontosaurids are thought to be widespread in the early Late Cretaceous of Asia, few such fossils have been recovered due to the poor rock record from that time. *Ulughbegsaurus* represents a previously unknown apex predator of the Turonian Bissekty Formation and the first reported carcharodontosaurian species from Late Cretaceous Central Asia. This taxon is one of the latest surviving Laurasian carcharodontosaurians and reveals the latest known stratigraphic co-occurrence between a carcharodontosaurian and a tyrannosauroid (i.e. *Timurlengia*). The latter suggests that carcharodontosaurians were still dominant predators in the Turonian, at least in Central Asia, while tyrannosauroids were smaller mesopredators. Because of the geographic location between East Asia and Europe, it remains uncertain if *Ulughbegsaurus* had affinities with European or East Asian/ Gondwanan carcharodontosaurians due to unresolved phylogenetic relationships. It is probable that fragmentary bones and isolated teeth from the Bissekty Formation of Uzbekistan may belong to carcharodontosaurians.

Ethics. The research was approved by the Ethics Review Committee of the Centre for Social Sciences (TK CSS). We confirm that all methods were carried out in accordance with relevant guidelines and regulations.

Data accessibility. The data are provided in the electronic supplementary material [55].

Authors' contributions. K.T. conceived of the study, designed the study, collected the data, analysed the data, carried out the statistical analyses and drafted the manuscript; O.U.O.A. participated in the design of the study, collected the data and drafted the manuscript; D.K.Z. participated in the design of the study, participated in data analysis and drafted the manuscript; A.S.A. coordinated the study and drafted the manuscript; Y.K. designed the study, collected the data and critically revised the manuscript. All authors gave final approval for publication and agree to be held accountable for the work performed therein.

Competing interests. We declare we have no competing interests.

Funding. This research was funded by Natural Sciences and Engineering Research Council of Canada with a project ID of 327513-2009 (to D.K.Z.).

Acknowledgement. We thank Qodirjon Ruzimovich Mingboyev, Sherzod Maxmudov and Ken-Ichiro Hisada for their support during the specimen observation at the State Geological Museum of the State Committee of the Republic of Uzbekistan on Geology and Mineral Resources. We are also grateful to Takanobu Tsuihiji, Makoto Manabe and Chisako Sakata for the specimen access at National Museum of Nature and Science, Tokyo, Japan. We appreciate Thomas Carr and an anonymous reviewer for their constructive advice.

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
