## [Peer Review File · Royal Society Open Science]

Review History

RSOS-210923.R0 (Original submission)

Review form: Reviewer 1 (Gregory Paul)

Is the manuscript scientifically sound in its present form?

No

Are the interpretations and conclusions justified by the results?

Yes

Is the language acceptable?

Yes

Do you have any ethical concerns with this paper?

No

Have you any concerns about statistical analyses in this paper?

No

Recommendation?

Major revision is needed (please make suggestions in comments)

Comments to the Author(s)

This is a useful, generally well done paper, but it has a particular serious issue that needs to be considered before publication, and probably requires a significant retraction. At the same time there is a potentially interesting issue that the authors should consider adding.

I see no particular issues with the structure, grammar, etc, although my editing skills in these areas are limited. The conclusion that the maxilla fragment belonged to a carcharodontosaur looks good, although I am not a leading expert on the issue, so see what other reviewer/s say on that. The subsequent conclusions that this is evidence that the Bissekty habitat was predator dominated by a carcharodontosaur over a smaller basal tyrannosauroid and associated conclusions about the evolution of these two groups in the Cretaceous are reasonable and scientifically interesting. Being a leading researcher on estimating body size in dinosaurs the estimates contained herein appear plausible. These items make the bulk of the paper worth publishing.

The big problem is using this small fragment of one bone as a holotype for a binomial name. I would not do it, and it is very dubious to do so. The specimen is the basis for yet another nomen dubium. It is not at all close to possible to use such a minor fossil item to begin to diagnose a species, and a genus is almost as problematic. I doubt this specimen could even be reliably assigned at the subfamily level. It is quite possible that more than one carcharodontosaur taxon was present in the Bissekty. If that proves true over time then the holotype will be of no use, and there will have to be an appeal to the ICZN to assign *Ulugbegsaurus uzbekistanensis* a better specimen as a new type if such a fossil is found, and so forth. This specimen as a holotype is about as bad as that for *Allosaurus fragilis*, which has required an ICZN petition to shift the name to a new type specimens. There will also be considerable confusion over other carcharodontosaurs from the same region and similar time frame. Even if this was a nearly complete maxilla with some teeth in place it would be a questionable holotype. I see that *Timurlengia* is based on just a braincase, but while this is not as bad, it is problematic. In my region I stopped the use of a complete large Early Cretaceous caudal from Washington DC being made into a holotype because although distinct from all other known contemporary theropod caudals, it suffered from the same defects noted above. Unfortunately the use of inadequate specimens as holotypes is continuing, and this practice needs to be stopped. The specimen does not meet current diagnostic standards for types, doing so does not add scientific value to the paper and reduces it including being a potential of future taxonomic trouble, and I strongly oppose this one aspect of the paper. The specimen should be designated *Carcharodontosauria incertae sedis*. The conclusions of the rest of the paper will still stand.

There is an issue I would like to see the authors address if possible. In many if not most late Mesozoic dinosaur habitats the top predators are very large. Is there information on why the largest known Bissekty theropod is only about a tonne? If this has been discussed elsewhere a note on the issue and a citation would be helpful, if not an original analysis would be productive, if such is possible with the data on hand.

I do not consider this paper publishable in its current form because the specimen should not be a holotype. It would like to see discussion of why the formation's theropods were not very big, but that is not critical. Otherwise I see no issues with the paper, and urge publication after the strongly recommended removal of the new genus and species name.

Review form: Reviewer 2 (Thomas Carr)

Is the manuscript scientifically sound in its present form?

Yes

Are the interpretations and conclusions justified by the results?

Yes

Is the language acceptable?

No

Do you have any ethical concerns with this paper?

No

Have you any concerns about statistical analyses in this paper?

No

Recommendation?

Major revision is needed (please make suggestions in comments)

Comments to the Author(s)

July 6, 2021

Dear Editor,

I submit to you my review of manuscript RSOS-210923, "A new carcharodontosaur theropod dinosaur occupies apex predator niche in the early Late Cretaceous of Uzbekistan" by Tanaka et al. The manuscript is a scientifically important contribution in that it documents a new taxon that illuminates the ecology of predatory theropods, analogues of which are seen elsewhere.

Although the ms is novel and interesting there are a few outstanding issues:

- 1) The usage of English requires a more thorough review by its native English author.
- 2) The informal usage of "carcharodontosaur" is jarring in that it is inconsistent with the other taxonomic names. For example, the informal "tyrannosauroid" is used in place of "Tyrannosauroida", hence the term "carcharodontosaurian" should be used since it is based on the formal clade name "Carcharodontosauria".
- 3) The use of a tyrannosauroid phylogeny to illustrate the several carcharodontosaurian + tyrannosauroid faunas is jarring, in that the article is about the former clade, not the latter.
- 4) Finally, the authors should have combined the two data matrices into a single matrix in order to recover a carcharodontosaurian phylogeny that exploits all of the available taxa instead of running two different matrices, which defeats the purpose of recovering the single Tree of Life. Brusatte and I took on that task for tyrannosauroids in 2016, which resolved conflicts between data sets that produced wildly different results.

The authors may know my identity: Thomas D. Carr.

Specific comments:

Title: change “carcharodontosaur” to “carcharodontosaurian” throughout the entire ms.

Abstract, line 51: delete “all told”, which is redundant.

Introduction, line 12: delete “the” since “Carcharodontosauria” is a proper name; insert “which” after the comma.

Line 25/26: delete “likely influenced” since you don’t say what the influence was.

Line 36: delete “of interest yet”, which adds nothing meaningful.

Lines 37/38: what are the names of the taxa?

Line 45: what is the “distant impact”? Unacceptably vague – fix.

Line 49: delete “theropod”.

Lines 51 to 53: non sequitur – fix.

Line 54: delete “Although...unidentified,”.

Page 4, line 6: change “represents one of” to “adds to”

Systematic palaeontology, lines 36 38: why this person? What’s the connection?

Line 48: delete “finger-like or”.

Line 49/50: delete “shaped” since “oval” is a shape; the word “prominences” is vague – be precise.

Description, line 12: change “a tooth in situ” to “an implanted tooth”.

Line 27: change “positioned” to “extending”; delete “visible” since you can see them.

Lines 31 to 33: the depressions are also present in the big tyrannosaurids.

Line 36: delete “finger-like”.

Line 42 delete “visible”; should “anterior” be changed to “ventral”?

Line 48: why is the fossa “considered here to be...maxillary fenestra”? Justify with evidence.

Line 52: “obscured laterally” by what?

Line 55: change “exhibits” to “has”.

Line 58: change “horizontal” to “horizontally”.

Page 6, line 7: change “side...process” to “bone” or “maxilla”.

Lines 7 to 10/11: this sentence is a mess – please fix.

Line 13: insert a comma after “below”.

Line 15: insert a comma after “plates”; replace “for” with “is seen in”.

Line 16/17: delete the specimen number; insert “together” after “fused”.

Line 19/20: insert a comma after “alveolus)”, change “becomes” to “becoming”, change “shallower” to “shallow”.

Line 27: delete “visible”.

Phylogenetic analysis, lines 3536 to 48/49: this para needs major surgery; please combine the matrices into one. I’m not submitting comments on the rest of this section except to say that it needs major trimming down.

Comments on isolated teeth, line 53/54: insert a comma after “general”.

Lines 55/56: change “exist” to “are seen”.

Line 58: delete “housed”; enclose “at UzSGM” in parentheses.

Page 8, line 13: change “elements” to “bones”.

Line 22: change “element” to “bone”; “thickness” is unacceptably vague – dorsoventral or mediolateral?

Line 24: delete “well-”.

Line 30: why not formalize the referral, why punt on it?

Line 33: insert “end of the” after “posterior” since there is no such thing as an “anterior maxilla” or a “posterior maxilla” – it is a single bone.

Discussion, line 7: change “appeared” to “were”.

Line 9: delete “heretofore”.

Line 16: change "major geographical" to "geographic"; delete "now".

Line 18: delete "via fossil evidence", delete "very".

Line 19: delete "appear to".

Line 22: change "migrated" to "dispersed".

Line 31: delete "reveals" and "the carcharodontosaur".

Line 33: delete "would have" – avoid passive voice.

Line 36: change "have been" to "are"; delete "previously".

Line 39: change "represent" to "are"; change "smaller-bodied" to "small", delete "than".

Line 40: delete "Ulugbegsaurus" and "known".

Line 42: insert "which is" after the comma.

Line 43: replace "appears" with "is".

Line 49: delete "-class".

Line 52: replace "ascertain" with "estimate"; replace "single elements" with "bones", insert "a" after the comma.

Line 54: delete "had to" change "partition" to "partitioned".

Line 55/56: delete "our discovery of".

Page 10, line 6: delete "probably".

Line 9: delete "known".

Line 15: insert "of" after "Cretaceous", delete "to our knowledge".

Line 16: delete the comma after "tyrannosauroid".

Line 18: delete "the...Timurlengia".

Line 19/20: change "gigantic" to "large"; change "diminutive" to "small".

Line 22/23: delete "interestingly".

Line 24: change "sympatric" to "sympatry"; replace "existence" to "between an"; insert "a" after "and".

Lines 27 to 28: delete "our...that" and "still...fairly".

Line 30: delete "at least" and the comma.

Lines 31/32: pluralize "allosauroid", delete "species".

Lines 33 to 34: this section makes no sense – please rewrite.

Line 43: delete "the" and "discovery".

Line 45: replace "appeared...been" with "were".

Line 46: replace "remains" with "was".

Conclusion, line 27: delete "to our knowledge".

Line 33: replace "remained" with "were".

Line 34: this section makes no sense – please fix.

Lines 37/38: replace "uncertain" to "small"; replace "prior to" with "before"; delete "importantly".

Figures: Please take the labels off the bones, and set them adjacent to the image.

Supp info:

Fig S3 and 4 captions: the use of the word "broad" is unclear
there are no references listed under "references"

Text S2: genus name "Yanchuanosaurus" is misspelled

Decision letter (RSOS-210923.R0)

Dear Dr Tanaka

The Editors assigned to your paper RSOS-210923 "A new carcharodontosaur theropod dinosaur occupies apex predator niche in the early Late Cretaceous of Uzbekistan" have now received comments from reviewers and would like you to revise the paper in accordance with the reviewer comments and any comments from the Editors. Please note this decision does not guarantee eventual acceptance.

Please submit your revised manuscript and required files (see below) no later than 21 days from today's (ie 08-Jul-2021) date. Note: the ScholarOne system will 'lock' if submission of the revision is attempted 21 or more days after the deadline. If you do not think you will be able to meet this deadline please contact the editorial office immediately.

on behalf of Prof Kevin Padian (Subject Editor)
openscience@royalsociety.org

Reviewer comments to Author:

Reviewer: 1

Comments to the Author(s)

This is a useful, generally well done paper, but it has a particular serious issue that needs to be considered before publication, and probably requires a significant retraction. At the same time there is a potentially interesting issue that the authors should consider adding.

I see no particular issues with the structure, grammar, etc, although my editing skills in these areas are limited. The conclusion that the maxilla fragment belonged to a carcharodontosaur looks good, although I am not a leading expert on the issue, so see what other reviewer/s say on that. The subsequent conclusions that this is evidence that the Bissekty habitat was predator dominated by a carcharodontosaur over a smaller basal tyrannosauroid and associated conclusions about the evolution of these two groups in the Cretaceous are reasonable and scientifically interesting. Being a leading researcher on estimating body size in dinosaurs the estimates contained herein appear plausible. These items make the bulk of the paper worth publishing.

The big problem is using this small fragment of one bone as a holotype for a binomial name. I would not do it, and it is very dubious to do so. The specimen is the basis for yet another nomen dubium. It is not at all close to possible to use such a minor fossil item to begin to diagnose a species, and a genus is almost as problematic. I doubt this specimen could even be reliably assigned at the subfamily level. It is quite possible that more than one carcharodontosaur taxon was present in the Bissekty. If that proves true over time then the holotype will be of no use, and there will have to be an appeal to the ICZN to assign *Ulugbegsaurus uzbekistanensis* a better specimen as a new type if such a fossil is found, and so forth. This specimen as a holotype is about as bad as that for *Allosaurus fragilis*, which has required an ICZN petition to shift the name to a new type specimens. There will also be considerable confusion over other carcharodontosaurs from the same region and similar time frame. Even if this was a nearly complete maxilla with some teeth in place it would be a questionable holotype. I see that *Timurlengia* is based on just a braincase, but while this is not as bad, it is problematic. In my region I stopped the use of a complete large Early Cretaceous caudal from Washington DC being made into a holotype because although distinct from all other known contemporary theropod caudals, it suffered from the same defects noted above. Unfortunately the use of inadequate specimens as holotypes is continuing, and this practice needs to be stopped. The specimen does not meet current diagnostic standards for types, doing so does not add scientific value to the paper and reduces it including being a potential of future taxonomic trouble, and I strongly oppose this one aspect of the paper. The specimen should be designated *Carcharodontosauria incertae sedis*. The conclusions of the rest of the paper will still stand.

There is an issue I would like to see the authors address if possible. In many if not most late Mesozoic dinosaur habitats the top predators are very large. Is there information on why the largest known Bissekty theropod is only about a tonne? If this has been discussed elsewhere a note on the issue and a citation would be helpful, if not an original analysis would be productive, if such is possible with the data on hand.

I do not consider this paper publishable in its current form because the specimen should not be a holotype. It would like to see discussion of why the formation's theropods were not very big, but that is not critical. Otherwise I see no issues with the paper, and urge publication after the strongly recommended removal of the new genus and species name.

Reviewer: 2
Comments to the Author(s)
July 6, 2021

Dear Editor,

I submit to you my review of manuscript RSOS-210923, "A new carcharodontosaur theropod dinosaur occupies apex predator niche in the early Late Cretaceous of Uzbekistan" by Tanaka et

al. The manuscript is a scientifically important contribution in that it documents a new taxon that illuminates the ecology of predatory theropods, analogues of which are seen elsewhere.

Although the ms is novel and interesting there are a few outstanding issues:

- 1) The usage of English requires a more thorough review by its native English author.
- 2) The informal usage of “carcharodontosaur” is jarring in that it is inconsistent with the other taxonomic names. For example, the informal “tyrannosauroid” is used in place of “Tyrannosauroida”, hence the term “carcharodontosaurian” should be used since it is based on the formal clade name “Carcharodontosauria”.
- 3) The use of a tyrannosauroid phylogeny to illustrate the several carcharodontosaurian + tyrannosauroid faunas is jarring, in that the article is about the former clade, not the latter.
- 4) Finally, the authors should have combined the two data matrices into a single matrix in order to recover a carcharodontosaurian phylogeny that exploits all of the available taxa instead of running two different matrices, which defeats the purpose of recovering the single Tree of Life. Brusatte and I took on that task for tyrannosauroids in 2016, which resolved conflicts between data sets that produced wildly different results.

The authors may know my identity: Thomas D. Carr.

Specific comments:

Title: change “carcharodontosaur” to “carcharodontosaurian” throughout the entire ms.

Abstract, line 51: delete “all told”, which is redundant.

Introduction, line 12: delete “the” since “Carcharodontosauria” is a proper name; insert “which” after the comma.

Line 25/26: delete “likely influenced” since you don’t say what the influence was.

Line 36: delete “of interest yet”, which adds nothing meaningful.

Lines 37/38: what are the names of the taxa?

Line 45” what is the “distant impact”? Unacceptably vague – fix.

Line 49: delete “theropod”.

Lines 51 to 53: non sequitur – fix.

Line 54: delete “Although...unidentified,”.

Page 4, line 6: change “represents one of” to “adds to”

Systematic palaeontology, lines 36 38: why this person? What’s the connection?

Line 48: delete “finger-like or”.

Line 49/50: delete “shaped” since “oval” is a shape; the word “prominences” is vague – be precise.

Description, line 12: change “a tooth in situ” to “an implanted tooth”.

Line 27: change “positioned” to “extending”; delete “visible” since you can see them.

Lines 31 to 33: the depressions are also present in the big tyrannosaurids.

Line 36: delete “finger-like”.

Line 42 delete “visible”; should “anterior” be changed to “ventral”?

Line 48: why is the fossa “considered here to be...maxillary fenestra”? Justify with evidence.

Line 52: “obscured laterally” by what?

Line 55: change “exhibits” to “has”.

Line 58: change “horizontal” to “horizontally”.

Page 6, line 7: change “side...process” to “bone” or “maxilla”.

Lines 7 to 10/11: this sentence is a mess – please fix.

Line 13: insert a comma after “below”.

Line 15: insert a comma after “plates”; replace “for” with “is seen in”.

Line 16/17: delete the specimen number; insert “together” after “fused”.

Line 19/20: insert a comma after “alveolus)”, change “becomes” to “becoming”, change “shallower” to “shallow”.

Line 27: delete “visible”.

Phylogenetic analysis, lines 3536 to 48/49: this para needs major surgery; please combine the matrices into one. I’m not submitting comments on the rest of this section except to say that it needs major trimming down.

Comments on isolated teeth, line 53/54: insert a comma after “general”.

Lines 55/56: change “exist” to “are seen”.

Line 58: delete “housed”; enclose “at UzSGM” in parentheses.

Page 8, line 13: change “elements” to “bones”.

Line 22: change “element” to “bone”; “thickness” is unacceptably vague – dorsoventral or mediolateral?

Line 24: delete “well-”.

Line 30: why not formalize the referral, why punt on it?

Line 33: insert “end of the” after “posterior” since there is no such thing as an “anterior maxilla” or a “posterior maxilla” – it is a single bone.

Discussion, line 7: change “appeared” to “were”.

Line 9: delete “heretofore”.

Line 16: change “major geographical” to “geographic”; delete “now”.

Line 18: delete “via fossil evidence”, delete “very”.

Line 19: delete “appear to”.

Line 22: change “migrated” to “dispersed”.

Line 31: delete “reveals” and “the carcharodontosaur”.

Line 33: delete “would have” – avoid passive voice.

Line 36: change “have been” to “are”; delete “previously”.

Line 39: change “represent” to “are”; change “smaller-bodied” to “small”, delete “than”.

Line 40: delete “Ulugbegsaurus” and “known”.

Line 42: insert “which is” after the comma.

Line 43: replace “appears” with “is”.

Line 49: delete “-class”.

Line 52: replace “ascertain” with “estimate”; replace “single elements” with “bones”, insert “a” after the comma.

Line 54: delete “had to” change “partition” to “partitioned”.

Line 55/56: delete “our discovery of”.

Page 10, line 6: delete “probably”.

Line 9: delete “known”.

Line 15: insert “of” after “Cretaceous”, delete “to our knowledge”.

Line 16: delete the comma after “tyrannosauroid”.

Line 18: delete “the...Timurlengia”.

Line 19/20: change “gigantic” to “large”; change “diminutive” to “small”.

Line 22/23: delete “interestingly”.

Line 24: change “sympatric” to “sympatry”; replace “existence” to “between an”; insert “a” after “and”.

Lines 27 to 28: delete “our...that” and “still...fairly”.

Line 30: delete “at least” and the comma.

Lines 31/32: pluralize "allosauroid", delete "species".
 Lines 33 to 34: this section makes no sense – please rewrite.
 Line 43: delete "the" and "discovery".
 Line 45: replace "appeared...been" with "were".
 Line 46: replace "remains" with "was".

Conclusion, line 27: delete "to our knowledge".
 Line 33: replace "remained" with "were".
 Line 34: this section makes no sense – please fix.
 Lines 37/38: replace "uncertain" to "small"; replace "prior to" with "before"; delete "importantly".

Figures: Please take the labels off the bones, and set them adjacent to the image.

Supp info:

Fig S3 and 4 captions: the use of the word "broad" is unclear
 there are no references listed under "references"
 Text S2: genus name "Yanchuanosaurus" is misspelled

===PREPARING YOUR MANUSCRIPT===

Your revised paper should include the changes requested by the referees and Editors of your manuscript. You should provide two versions of this manuscript and both versions must be provided in an editable format:
 one version identifying all the changes that have been made (for instance, in coloured highlight, in bold text, or tracked changes);
 a 'clean' version of the new manuscript that incorporates the changes made, but does not highlight them. This version will be used for typesetting if your manuscript is accepted.

===PREPARING YOUR REVISION IN SCHOLARONE===

Author's Response to Decision Letter for (RSOS-210923.R0)

See Appendix A.

RSOS-210923.R1 (Revision)

Review form: Reviewer 1 (Gregory Paul)

Is the manuscript scientifically sound in its present form?

Yes

Are the interpretations and conclusions justified by the results?

Yes

Is the language acceptable?

Yes

Do you have any ethical concerns with this paper?

No

Have you any concerns about statistical analyses in this paper?

No

Recommendation?

Accept as is

Comments to the Author(s)

No additional comments beyond those in the original review

Review form: Reviewer 2 (Thomas Carr)

Is the manuscript scientifically sound in its present form?

Yes

Are the interpretations and conclusions justified by the results?

Yes

Is the language acceptable?

Yes

Do you have any ethical concerns with this paper?

No

Have you any concerns about statistical analyses in this paper?

No

Recommendation?

Accept with minor revision (please list in comments)

Comments to the Author(s)

August 9, 2021

Dear editor,

I present to you my review of the revised draft of "A new carcharodontosaurian theropod dinosaur occupies apex predator niche in the early Late Cretaceous of Uzbekistan." In terms of readability, this draft is a great improvement over the first submission, and I applaud the authors for that. Although not all of my suggestions were followed (see below), I do think the ms is in shape for publication, aside from two minor revisions:

Page 4, Locality and horizon: "...the precise location is unknown." Can a brief sentence be added to explain why this is so? Also, the authors should include a sentence that states that the fossil was collected by their museum.

Page 7, Phylogenetic analysis: please change "performed" to "done."

I do agree with Reviewer 1 that naming a fragment such as this with a Linnean binomial is problematic; in fact, my first impression was skeptical, but I am persuaded by the diagnosis of the new taxon that a name is appropriate. However, as implied by Reviewer 1, it may turn out that the diagnostic characters may turn out to have a wider distribution among the clade, leaving the new taxon without empirical support. But time will tell.

I accept the authors' reasons for not combining the two matrices - it's a big job to take on. However, this team could have been the first to take on the task, thereby increasing the scientific value of the article (which was my motivation for making the suggestion).

In the end, I think that this article is an important contribution in that it clarifies the diversity of theropods in the early Late Cretaceous in Central Asia and sharpens the regional snapshot of diversity across Laurasia. The formal documentation of sympatric carcharodontosaurians and tyrannosauroids in Uzbekistan justifies publication, whether or not the maxilla is diagnostic to species. Otherwise, the Bissekty fauna would stand out - inaccurately - as an outlier among the Late Jurassic to Late Cretaceous ecosystems of Laurasia where sympatry is also seen. This discovery also figures into the global terrestrial ecological picture where the turnover from one dominant theropod clade to another is seen, which extends this study's importance beyond the immediate issue of allosauroid to tyrannosaurid turnover that was completed in the Late Cretaceous of Laurasia back to, arguably, the Early Jurassic.

The authors may know my identity: Thomas D. Carr.

Sincerely,
Thomas D. Carr, PhD
Associate Professor
Department of Biology
Carthage College
2001 Alford Park Drive
Kenosha, WI

Decision letter (RSOS-210923.R1)

Dear Dr Tanaka

On behalf of the Editors, we are pleased to inform you that your Manuscript RSOS-210923.R1 "A new carcharodontosaurian theropod dinosaur occupies apex predator niche in the early Late Cretaceous of Uzbekistan" has been accepted for publication in Royal Society Open Science subject to minor revision in accordance with the referees' reports. Please find the referees' comments along with any feedback from the Editors below my signature.

Please submit your revised manuscript and required files (see below) no later than 7 days from today's (ie 16-Aug-2021) date. Note: the ScholarOne system will 'lock' if submission of the revision is attempted 7 or more days after the deadline. If you do not think you will be able to meet this deadline please contact the editorial office immediately.

on behalf of Professor Kevin Padian (Subject Editor)
openscience@royalsociety.org

Associate Editor Comments to Author:

Overall, the reviewers are satisfied with your revision, though a few minor comments remain that you are encouraged to respond to and incorporate the reviewers' final suggestions. Many thanks for your support.

Reviewer comments to Author:

Reviewer: 1

Comments to the Author(s)

No additional comments beyond those in the original review

Reviewer: 2
Comments to the Author(s)

August 9, 2021

Dear editor,

I present to you my review of the revised draft of "A new carcharodontosaurian theropod dinosaur occupies apex predator niche in the early Late Cretaceous of Uzbekistan." In terms of readability, this draft is a great improvement over the first submission, and I applaud the authors for that. Although not all of my suggestions were followed (see below), I do think the ms is in shape for publication, aside from two minor revisions:

Page 4, Locality and horizon: "...the precise location is unknown." Can a brief sentence be added to explain why this is so? Also, the authors should include a sentence that states that the fossil was collected by their museum.

Page 7, Phylogenetic analysis: please change "performed" to "done."

I do agree with Reviewer 1 that naming a fragment such as this with a Linnean binomial is problematic; in fact, my first impression was skeptical, but I am persuaded by the diagnosis of the new taxon that a name is appropriate. However, as implied by Reviewer 1, it may turn out that the diagnostic characters may turn out to have a wider distribution among the clade, leaving the new taxon without empirical support. But time will tell.

I accept the authors' reasons for not combining the two matrices - it's a big job to take on. However, this team could have been the first to take on the task, thereby increasing the scientific value of the article (which was my motivation for making the suggestion).

In the end, I think that this article is an important contribution in that it clarifies the diversity of theropods in the early Late Cretaceous in Central Asia and sharpens the regional snapshot of diversity across Laurasia. The formal documentation of sympatric carcharodontosaurians and tyrannosauroids in Uzbekistan justifies publication, whether or not the maxilla is diagnostic to species. Otherwise, the Bissekty fauna would stand out - inaccurately - as an outlier among the Late Jurassic to Late Cretaceous ecosystems of Laurasia where sympatry is also seen. This discovery also figures into the global terrestrial ecological picture where the turnover from one dominant theropod clade to another is seen, which extends this study's importance beyond the immediate issue of allosauroid to tyrannosaurid turnover that was completed in the Late Cretaceous of Laurasia back to, arguably, the Early Jurassic.

The authors may know my identity: Thomas D. Carr.

Sincerely,
Thomas D. Carr, PhD
Associate Professor
Department of Biology
Carthage College
2001 Alford Park Drive
Kenosha, WI

===PREPARING YOUR MANUSCRIPT===

===PREPARING YOUR REVISION IN SCHOLARONE===

Author's Response to Decision Letter for (RSOS-210923.R1)

See Appendix B.

Decision letter (RSOS-210923.R2)

Dear Dr Tanaka,

I am pleased to inform you that your manuscript entitled "A new carcharodontosaurian theropod dinosaur occupies apex predator niche in the early Late Cretaceous of Uzbekistan" is now accepted for publication in Royal Society Open Science.

on behalf of Kevin Padian (Subject Editor)
openscience@royalsociety.org

Appendix A

Response to the reviewers

Thank you for managing our manuscript. We appreciate all constructive and helpful advice/suggestions from the reviewers. We carefully revised our manuscript according to their suggestions as much as possible. Below are our responses highlighted in blue.

Reviewer: 1

The big problem is using this small fragment of one bone as a holotype for a binomial name. I would not do it, and it is very dubious to do so. The specimen is the basis for yet another nomen dubium. It is not at all close to possible to use such a minor fossil item to begin to diagnose a species, and a genus is almost as problematic. I doubt this specimen could even be reliably assigned at the subfamily level. It is quite possible that more than one carcharodontosaur taxon was present in the Bissekty. If that proves true over time then the holotype will be of no use, and there will have to be an appeal to the ICZN to assign *Ulugbegsaurus uzbekistanensis* a better specimen as a new type if such a fossil is found, and so forth. This specimen as a holotype is about as bad as that for *Allosaurus fragilis*, which has required an ICZN petition to shift the name to a new type specimens. There will also be considerable confusion over other carcharodontosaurs from the same region and similar time frame. Even if this was a nearly complete maxilla with some teeth in place it would be a questionable holotype. I see that *Timurlengia* is based on just a braincase, but while this is not as bad, it is problematic. In my region I stopped the use of a complete large Early Cretaceous caudal from Washington DC being made into a holotype because although distinct from all other known contemporary theropod caudals, it suffered from the same defects noted above.

Unfortunately the use of inadequate specimens as holotypes is continuing, and this practice needs to be stopped. The specimen does not meet current diagnostic standards for types, doing so does not add scientific value to the paper and reduces it including being a potential of future taxonomic trouble, and I strongly oppose this one aspect of the paper. The specimen should be designated *Carcharodontosauria incertae sedis*. The conclusions of the rest of the paper will still stand.

We appreciate this thoughtful advice and reconsidered our decision to name a new dinosaur species based on the described fossil specimen. However, we believe we have more than enough evidence to allow us to assign this maxillary specimen to a new genus/species.

First, the maxilla is among the most diagnostic bones in the theropod skeleton, and generally contains many important taxonomic characteristics. For example, Hendrickx and Mateus (2014) indicate that morphology of maxilla is highly variable among theropods and “this bone provides far more information than many other parts of the skeleton, and the diagnostic value of the maxilla is significant”. Thus, the maxilla is often used to taxonomically identify and diagnose specimens. Among 13 carcharodontosaurian species for which the maxilla is known, 77% of the taxa (= 10 species) possess traits from the maxilla in their taxonomic diagnosis. We found four diagnostic characteristics in our specimen. In other words, the number of bones used to diagnose or name a new species is not important, but rather it is the amount of taxonomic information present in a given bone or bones.

Second, the validity of *Ulughbegsaurus* is also supported by phylogenetic analyses. We performed two separate analyses based on two different data matrices and obtained consistent results in that *Ulughbegsaurus* is placed within Carcharodontosauria for both. The maxilla is known from most members of this Carcharodontosauria (>70%) and the maxilla of *Ulughbegsaurus* clearly differs from them and has its own unique characteristics. These unique characters (autapomorphies) and the phylogenetic analyses are both consistent with this fossil being a new and distinct taxon.

Third, the reviewer is concerned of the possibility that a new carcharodontosaur specimen could be found from the Bissekty Formation in the future, and then “the holotype will be of no use”. We argue it will be of use because a holotype specimen, including that of *Ulughbegsaurus*, must have autapomorphies, and if a future specimen has these features, then it can be assigned to the already known species. If not, that specimen may be a new species or may be not be assignable to a species if it lacks unique characters. The diagnosis of a species can be expanded if additional specimens or bones are found, a common practice in paleontology. We do not think an ICZN petition will be relevant to the naming of this species.

Fourth, many species are actually named based on a single bone, including the maxilla (e.g., *Carcharodontosaurus iguidensis*). This is true in many carcharodontosaur taxa, such as *Mapusaurus*, *Veterupristisaurus*, *Eocarcharia*, *Taurovenator*, and *Sauroniops*. Furthermore, all recovered dinosaur materials are disarticulated in the Bissekty Formation and all named dinosaur taxa from the formation are based on a single bone/element: an incomplete maxilla for *Turanoceratops* (Sues and Averianov, 2009b), a caudal vertebra for *Dzharatitanis* (Averianov and Sues, 2021), dentaries for *Urbacodon* and *Caenagnathasia martinsoni* (Averianov and Sues, 2007; Sues and Averianov, 2015), a braincase for *Bissektipelta*, *Timurlengia*, and *Itemirus* (Averianov,

2002; Brusatte et al., 2006; Sues and Averianov, 2014), a partial skull roof for *Levnesovia* (Sues and Averianov, 2009a), and an isolated tooth for *Paronychodon* and *Richardoestesia asiatica* (Sues and Averianov, 2013).

Due to all the reasons stated above, even though our specimen is an isolated maxilla, it is appropriate to establish a new taxon.

There is an issue I would like to see the authors address if possible. In many if not most late Mesozoic dinosaur habitats the top predators are very large. Is there information on why the largest known Bissekty theropod is only about a tonne? If this has been discussed elsewhere a note on the issue and a citation would be helpful, if not an original analysis would be productive, if such is possible with the data on hand. This is an intriguing question. In Table S2, we compiled body mass of top predators from 50 localities. Among them, mega-predators (much larger than 1000 kg) are present in 30 localities, while 14 localities lack mega-predators. The remaining six localities are unknown due to incomplete discoveries of predatory dinosaurs. These 14 localities tend to be older than late Late Cretaceous and/or to be poorly studied areas. Similar results are obtained by Schroeder et al. (2021) and Holtz (in press) who found mega-predators from well-studied formations. One reason why mega-predators are absent in the Bissekty Formation could be related to poor sampling or preservational biases. However, we could not find any previous papers that discuss the absence of mega-predators in these formations and thus we find it very difficult to include this in the discussion of the manuscript.

Reviewer: 2

1) The usage of English requires a more thorough review by its native English author. The manuscript was double-checked again by the native English author (DKZ).

2) The informal usage of “carcharodontosaur” is jarring in that it is inconsistent with the other taxonomic names. For example, the informal “tyrannosauroid” is used in place of “Tyrannosauroidea”, hence the term “carcharodontosaurian” should be used since it is based on the formal clade name “Carcharodontosauria”.

We modified “carcharodontosaur” to “carcharodontosaurian” throughout the manuscript according to the reviewer’s comment.

3) The use of a tyrannosauroid phylogeny to illustrate the several carcharodontosaurian + tyrannosauroid faunas is jarring, in that the article is about the former clade, not the latter.

We appreciate the reviewer's point and have considered such an alternative option. However, the purpose of Figure 4 is to show: 1) tyrannosauroid species were relatively small in the formations where Allosauroidea co-occurred; and 2) the appearance of large-bodied tyrannosaurids occurred in the Campanian, after the disappearance of Allosauroidea in Laurasia. Due to these purposes, it is better to use a tyrannosauroid phylogeny with co-occurring allosauroidea taxa. Otherwise (i.e., by showing an allosauroidea phylogeny), we cannot express our discussion in the figure.

4) Finally, the authors should have combined the two data matrices into a single matrix in order to recover a carcharodontosaurian phylogeny that exploits all of the available taxa instead of running two different matrices, which defeats the purpose of recovering the single Tree of Life. Brusatte and I took on that task for tyrannosauroids in 2016, which resolved conflicts between data sets that produced wildly different results.

We much appreciate the reviewer's advice, but this suggestion would be beyond the scope of our research. The purpose of the phylogenetic analyses in this study is to identify the (approximate) position of our specimen, not to reveal the precise interrelationships among Carcharodontosauria. We obtained a consistent result from both analyses in that our specimen is placed in Carcharodontosauria. Both of the matrices used for our phylogenetic analyses are recent, valid and published. We chose to run both in order to double-check the phylogenetic position of our taxon as Carcharodontosauria, which did not change, and decided to present both in the manuscript. We could have run and presented only the most recent matrix for the manuscript, which many (probably most) authors do. The importance here is that, even if these two matrices show different phylogenetic positions for some other taxa, our specimen is placed within Carcharodontosauria. Thus, we would like to use the original analyses in this study.

Specific comments:

Title: change "carcharodontosaur" to "carcharodontosaurian" throughout the entire ms. We modified the term throughout the entire ms according to the comment.

Abstract, line 51: delete “all told”, which is redundant.

We deleted “all told” in Abstract.

Introduction, line 12: delete “the” since “Carcharodontosauria” is a proper name; insert “which” after the comma.

We deleted “the” and inserted “which” according to the comment.

Line 25/26: delete “likely influenced” since you don’t say what the influence was.

We replaced “likely influenced” to “is likely related to”.

Line 36: delete “of interest yet”, which adds nothing meaningful.

We deleted “of interest yet”.

Lines 37/38: what are the names of the taxa?

We inserted “(*Siats* and *Moros*, respectively)”.

Line 45” what is the “distant impact”? Unacceptably vague – fix.

We modified the later half of the sentence, “, a time period well before the ascent of tyrannosauroids”.

Line 49: delete “theropod”.

We deleted “theropod”.

Lines 51 to 53: non sequitur – fix.

We separated the sentence into two.

Line 54: delete “Although...unidentified,”.

We deleted “Although...unidentified,”.

Page 4, line 6: change “represents one of” to “adds to”

We replaced “represents one of” to “adds to”.

Systematic palaeontology, lines 36 38: why this person? What’s the connection?

We revised Etymology; ‘*Ulughbeg*’ refers to Ulugh Beg, in recognition of his early scientific contributions as a fifteenth century astronomer, mathematician and Timurid

sultan in central Asia region (now Uzbekistan). ‘*Sauros*’ meaning reptiles in Latin. Specific name, ‘*uzbekistan*’, refers to the Republic of Uzbekistan.

Line 48: delete “finger-like or”.

We deleted “finger-like or”.

Line 49/50: delete “shaped” since “oval” is a shape; the word “prominences” is vague – be precise.

We deleted “shaped” and replaced “prominences” to “tubercles” (throughout the manuscript).

Description, line 12: change “a tooth in situ” to “an implanted tooth”.

We replaced “a tooth in situ” to “an implanted tooth”.

Line 27: change “positioned” to “extending”; delete “visible” since you can see them.

We replaced “positioned” to “extending” and “visible” to “situated”.

Lines 31 to 33: the depressions are also present in the big tyrannosaurids.

We inserted “similar to some large tyrannosaurids (figure 2A [36, 37])” after “shallow depressions that are often bounded by a ridge”.

Line 36: delete “finger-like”.

We deleted “finger-like or” and “shaped”.

Line 42 delete “visible”; should “anterior” be changed to “ventral”?

We replaced “visible” to “located” and “anterior” to “anteroventral”.

Line 48: why is the fossa “considered here to be...maxillary fenestra”? Justify with evidence.

We modified the sentence to “This accessory fossa is considered here to be an extension of the maxillary fenestra due to its adjacent location.”.

Line 52: “obscured laterally” by what?

We inserted “by the rim of the antorbital fossa” after “...is obscured laterally”.

Line 55: change “exhibits” to “has”.

We replaced “exhibits” to “has”.

Line 58: change “horizontal” to “horizontally”.

We replaced “horizontal” to “horizontally”.

Page 6, line 7: change “side...process” to “bone” or “maxilla”.

We replaced “side...process” to “bone”.

Lines 7 to 10/11: this sentence is a mess – please fix.

We separated the sentence into two: “The dorsal surface of the process to the apex of the main body represents a contact surface for the nasal. This surface is flat, smooth, and narrows posteriorly.”.

Line 13: insert a comma after “below”.

We inserted a comma.

Line 15: insert a comma after “plates”; replace “for” with “is seen in”.

We inserted a comma after “plates” and replaced “for” with “is seen in”.

Line 16/17: delete the specimen number; insert “together” after “fused”.

We deleted the specimen number and inserted “together” after “fused”.

Line 19/20: insert a comma after “alveolus)”, change “becomes” to “becoming”, change “shallower” to “shallow”.

We inserted a comma after “alveolus)”, replaced “becomes” to “becoming”, and replaced “shallower” to “shallow”.

Line 27: delete “visible”.

We replaced “visible” to “remained”.

Phylogenetic analysis, lines 35/36 to 48/49: this para needs major surgery; please combine the matrices into one. I’m not submitting comments on the rest of this section except to say that it needs major trimming down.

Please see the above response for “4) Finally, the authors should have combined the two data...”.

Comments on isolated teeth, line 53/54: insert a comma after “general”.

We inserted a comma after “general”.

Lines 55/56: change “exist” to “are seen”.

We replaced “exist” to “are seen”.

Line 58: delete “housed”; enclose “at UzSGM” in parentheses.

We deleted “, housed” and enclosed “at UzSGM” in parentheses.

Page 8, line 13: change “elements” to “bones”.

We replaced “elements” to “bones”.

Line 22: change “element” to “bone”; “thickness” is unacceptably vague – dorsoventral or mediolateral?

We replaced “elements” to “bones” and inserted “mediolateral” before “thickness”.

According to the reviewer’s comment, Text S2 of Supporting Information was also fixed (“the dorsoventral height and the mediolateral thickness of the jugal ramus”).

Line 24: delete “well-“.

We deleted “well-“.

Line 30: why not formalize the referral, why punt on it?

We chose the specimen as a referred specimen and added a section of ‘Referred specimens’ in Systematic Palaeontology.

Line 33: insert “end of the” after “posterior” since there is no such thing as an “anterior maxilla” or a “posterior maxilla” – it is a single bone.

We inserted “end of the” after “posterior”.

Discussion, line 7: change “appeared” to “were”.

We replaced “appeared” to “were”.

Line 9: delete “heretofore”.

We deleted “heretofore”.

Line 16: change “major geographical” to “geographic”; delete “now”.
We replaced “major geographical” to “geographic” and deleted “now”.

Line 18: delete “via fossil evidence”, delete “very”.
We deleted “via fossil evidence” and “very”.

Line 19: delete “appear to”.
We deleted “appear to”.

Line 22: change “migrated” to “dispersed”.
We replaced “migrated” to “dispersed”.

Line 31: delete “reveals” and “the carcharodontosaur”.
We deleted “reveals that the carcharodontosaur”.

Line 33: delete “would have” – avoid passive voice.
We deleted “would have”.

Line 36: change “have been” to “are”; delete “previously”.
We replaced “have been” to “are” and deleted “previously”.

Line 39: change “represent” to “are”; change “smaller-bodied” to “small”, delete “than”.
We replaced “represent” to “are” and “smaller-bodied” to “smaller forms”.

Line 40: delete “Ulugbegsaurus” and “known”.
We deleted “Ulugbegsaurus” and “known”.

Line 42: insert “which is” after the comma.
We inserted “which is” after the comma.

Line 43: replace “appears” with “is”.
We replaced “appears” with “is”.

Line 49: delete “-class”.
We deleted “-class”.

Line 52: replace “ascertain” with “estimate”; replace “single elements” with “bones”, insert “a” after the comma.

We replaced “ascertain” with “estimate” and “single elements” with “bones”, and inserted “a” after the comma.

Line 54: delete “had to” change “partition” to “partitioned”.

We deleted “had to” and replaced “partition” to “partitioned”.

Line 55/56: delete “our discovery of”.

We deleted “our discovery of”.

Page 10, line 6: delete “probably”.

We deleted “probably”.

Line 9: delete “known”.

We deleted “known”.

Line 15: insert “of” after “Cretaceous”, delete “to our knowledge”.

We inserted “of” after “Cretaceous” and deleted “, to our knowledge,”.

Line 16: delete the comma after “tyrannosauroid”.

We deleted the comma after “tyrannosauroid”.

Line 18: delete “the... Timurlengia”.

We deleted “the... Timurlengia”.

Line 19/20: change “gigantic” to “large”; change “diminutive” to “small”.

We replaced “gigantic” to “large” and “diminutive” to “small”.

Line 22/23: delete “interestingly”.

We deleted “interestingly,”.

Line 24: change “sympatric” to “sympatry”; replace “existence” to “between an”; insert “a” after “and”.

We replaced “sympatric” to “sympatry” and “existence” to “between an”, and inserted

“a” after “and”.

Lines 27 to 28: delete “our...that” and “still...fairly”.

We deleted “our...that” and “still...fairly”.

Line 30: delete “at least” and the comma.

We deleted “at least” and the comma.

Lines 31/32: pluralize “allosauroid”, delete “species”.

We pluralized “allosauroid” and deleted “species”.

Lines 33 to 34: this section makes no sense – please rewrite.

We modified the section to “even in formations where tyrannosauroid species co-occurred.”.

Line 43: delete “the” and “discovery”.

We deleted “the” and “discovery”.

Line 45: replace “appeared...been” with “were”.

We replaced “appeared...been” with “were”.

Line 46: replace “remains” with “was”.

We replaced “remains” with “was”.

Conclusion, line 27: delete “to our knowledge”.

We deleted “to our knowledge”.

Line 33: replace “remained” with “were”.

We replaced “remained” with “were”.

Line 34: this section makes no sense – please fix.

We realized the sentence pointed out by the reviewer is redundant and thus it was deleted.

Lines 37/38: replace “uncertain” to “small”; replace “prior to” with “before”; delete “importantly”.

We replaced “uncertain” to “small” and “prior to” with “before”, and deleted “importantly”.

Figures: Please take the labels off the bones, and set them adjacent to the image.
We revised Figs. 1 and S5 according to the comment.

Supp info:

Fig S3 and 4 captions: the use of the word "broad" is unclear

We deleted “broad” from the captions.

there are no references listed under "references"

We left “References” because it is placed in the table of contents and the reference list is not needed here.

Text S2: genus name "Yanchuanosaurus" is misspelled

We fixed it to “*Yangchuanosaurus*”.

Other changes.

1) We replaced the genus name to “*Ulughbegasaurus*” throughout the manuscript.
Zoobank was also revised.

2) We added the section of ‘Institutional abbreviations’ at the end of Introduction.
Institutional abbreviations: CCMGE, Chernyshev’s Central Museum of Geological Exploration, Saint Petersburg, Russia; UzSGM, State Geological Museum of the State Committee of the Republic of Uzbekistan on Geology and Mineral Resources; ZIN PH, Paleontological Collection, Zoological Institute, Russian Academy of Science, Saint Petersburg, Russia.

3) We added the section of referred specimens. ‘Referred specimens: CCMGE 600/12457 is a jugal ramus of a left maxilla and ZIN PH 357/16 is a posterior end of the right maxilla, previously referred to the dromaeosaurid *Itemirus medullaris* ([18]; see ‘5. Comments on isolated teeth and bone fragments of theropods from the Bissekty Formation’) after the ‘Holotype’ section in ‘2. Systematic palaeontology’.

- 4) We deleted two sentences in the last paragraph of Discussion as we realized it is not necessary in this study, “The tyrannosauroid found from the Baynshiree Formation was initially reported as *Alectrosaurus* [54], but its assignment is questioned by some studies [55,56]. If the Baynshiree taxon is equivalent in size to *Alectrosaurus* (627 kg [3]), then the tyrannosauroid is clearly larger than *Timurlengia* (172 kg, table S2)”.
- 5) We added a sentence in the Acknowledgement, “We appreciate Thomas Carr and an anonymous reviewer for their constructive advice”.

Appendix B

Response to the reviewers

Thank you again for managing our manuscript. We are glad that the reviewers overall satisfied our revised manuscript. Also, we appreciate two additional constructive suggestions from the reviewer 2 (Dr. Thomas Carr). We revised our manuscript according to his suggestions. Below are our responses highlighted in blue.

Reviewer: 2

Comments to the Author(s)

Page 4, Locality and horizon: "...the precise location is unknown." Can a brief sentence be added to explain why this is so? Also, the authors should include a sentence that states that the fossil was collected by their museum.

We modified the sentence to '...the precise location of the holotype is unknown as geographic coordinates were not noted in the field'. Also, we added a sentence: 'The holotype was brought to UzSGM by a field team member of Lev Alexandrovich Nesson'.

Page 7, Phylogenetic analysis: please change "performed" to "done."

We replaced 'performed' to 'done' in L9-10 of page 10.